# Identification of *Plasmodium falciparum* circumsporozoite protein-specific CD8+ T cell epitopes in a malaria exposed population

**Kwadwo A. Kusi**[1,2]*, **Felix E. Aggor**[2¤], **Linda E. Amoah**[1,2], **Dorothy Anum**[1], **Yvonne Nartey**[3], **Daniel Amoako-Sakyi**[3], **Dorcas Obiri-Yeboah**[3], **Michael Hollingdale**[4], **Harini Ganeshan**[4], **Maria Belmonte**[4], **Bjoern Peters**[5], **Yohan Kim**[5], **John Tetteh**[1], **Eric Kyei-Baafour**[1], **Daniel Dodoo**[1], **Eileen Villasante**[4], **Martha Sedegah**[4]

**1** Department of Immunology, Noguchi Memorial Institute for Medical Research, College of Health Sciences, University of Ghana, Legon, Accra, Ghana, **2** Department of Biochemistry, Cell and Molecular Biology, College of Basic and Applied Sciences, University of Ghana, Legon, Accra, Ghana, **3** Department of Microbiology and Immunology, College of Health and Allied Sciences, University of Cape Coast, Cape Coast, Ghana, **4** Malaria Department, Naval Medical Research Center, Silver Springs, MD, United States of America, **5** La Jolla Institute for Immunology, La Jolla, CA, United States of America

¤ Current address: Department of Immunology, University of Pittsburgh School of Medicine, Pittsburgh, PA, United States of America

* akusi@noguchi.ug.edu.gh

**Data Availability Statement:** All relevant data are within the manuscript and its Supporting Information files.

## Abstract

### Background

Sterile protection against malaria, most likely mediated by parasite-specific CD8+ T cells, has been achieved by attenuated sporozoite vaccination of animals as well as malaria-naïve and malaria-exposed subjects. The circumsporozoite protein (CSP)-based vaccine, RTS,S, shows low efficacy partly due to limited CD8+ T cell induction, and inclusion of such epitopes could improve RTS,S. This study assessed 8-10mer CSP peptide epitopes, present in predicted or previously positive *P. falciparum* 3D7 CSP 15mer overlapping peptide pools, for their ability to induce CD8+ T cell IFN-γ responses in natural malaria-exposed subjects.

### Methods

Cryopreserved PBMCs from nine HLA-typed subjects were stimulated with 23 8-10mer CSP peptides from the 3D7 parasite in IFN-γ ELISpot assays. The CD8+ T cell specificity of IFN-γ responses was confirmed in ELISpot assays using CD8+ T cell-enriched PBMC fractions after CD4+ cell depletion.

### Results

Ten of 23 peptide epitopes elicited responses in whole PBMCs from five of the nine subjects. Four peptides tested positive in CD8+ T cell-enriched PBMCs from two previously positive responders and one new subject. All four immunodominant peptides are restricted by globally common HLA supertypes (A02, A03, B07) and mapped to regions of the CSP

**Funding:** This study was supported by a University of Ghana Research Fund Grant number URF/6/ILG-003/2012-2013, awarded to KAK, and with resources from the Naval Medical Research Center (work unit number 6000.RAD1.F.A0309). The views expressed herein are the personal ones of the authors and do not purport to reflect the views of the US Navy or the Department of Defense. The funders had no role in study design, data collection and analysis, decision to publish, or preparation of the manuscript.

**Competing interests:** I have read the journal's policy and the authors of this manuscript have the following competing interests: The authors declare that they have no competing interests.

antigen with limited or no reported polymorphism. Association of these peptide-specific responses with anti-malarial protection remains to be confirmed.

## Conclusions

The relatively conserved nature of the four identified epitopes and their binding to globally common HLA supertypes makes them good candidates for inclusion in potential multi-epitope malaria vaccines.

## Background

Protective sterilizing immunity against malaria has been achieved in malaria-naïve humans following immunization with attenuated merozoites [1,2], irradiated *P. falciparum* sporozoites [3,4] or with live sporozoites under chloroquine prophylaxis before the establishment of blood stage infection [5,6]. Though there are no clearly defined correlates of protection against clinical malaria by these vaccines, immune mechanisms mediating protection may include interferon-γ (IFN-γ)-secreting CD8+ T cells that primarily target malaria antigens expressed on the surface of hepatocytes [7–9]. Despite the near 100% sterile protection achieved by these vaccines against homologous parasite strains, recent evidence suggest that whole sporozoite vaccines may have to include sporozoites from multiple parasite strains to induce long-term broad protection [10,11]. This approach however introduces new challenges with increased production cost and appropriate dosing.

An alternative approach to whole sporozoite immunization is to identify immunodominant epitopes within essential target parasite antigens for the development of multi-epitope subunit vaccines. This vaccine design only requires the production and formulation of short linear peptides or their corresponding DNA sequences, enabling multiple antigens from different parasite strains to be included in a single vaccine. In addition, relatively lower doses will be required for the induction of optimal protective responses [12,13]. Such immunodominant HLA-restricted T cell peptides from essential parasite antigens have been identified in immunized malaria-naïve individuals [14–16]. Initial assessment of vaccines designed on this basis have shown promising results [17–19].

Induction of sporozoite-specific CD8+ T cell responses requires the processing and presentation of sporozoite antigen peptides on infected hepatocytes via HLA class I molecules [20]. The genetic diversity within human HLA molecules could present a challenge to the development of broadly effective epitope-based vaccines. This can however be overcome by targeting parasite peptides that can be recognized and presented by multiple HLA class I supertypes [21].

Circumsporozoite protein (CSP) is the most abundant protein expressed on the surface of *Plasmodium* sporozoites and plays a crucial role in the invasion of hepatocytes [22]. It is the parasite component of RTS,S, the most advanced malaria vaccine. Previous studies with nine pools of 15mer overlapping peptides covering the entire 3D7 strain CSP antigen identified four pools (Cp1, Cp4, Cp6, Cp9) that induced positive IFN-γ responses among Ghanaian adult subjects with a history of *Plasmodium* infections over their life time [23,24]. Since each of these pools contained multiple 15mer peptides, the next step is to determine the minimal (8-10mer) epitope(s) that are ultimately responsible for the observed positive pool-specific responses.

The aim of this study was to experimentally assess the induction and T cell subset-specificity of IFN-γ responses by selected 8-10mer single peptides from *P. falciparum* CSP using

PBMCs from HLA-typed subjects with natural exposure to malaria. The selected peptides have been predicted by bioinformatics algorithms to bind to defined HLA types and/or were present in peptide pools that previously tested positive in ELISpot assays [23,24]. Overall, we identified four HLA-promiscuous and relatively conserved peptides that induce CD8+ T cell-specific IFN-γ responses.

## Methods

### Ethics

This study was conducted at the Noguchi Memorial Institute for Medical Research (NMIMR) according to a human research protocol that was approved by the NMIMR Institutional Review Board (Protocol number 042/13-14). The NMIMR-IRB holds a US Government Federal-wide Assurance (FWAA00001824) from the US Office for Human Research Protections. Written informed consent was sought from all study subjects who willingly agreed to be part of the study and met the inclusion criteria. All study procedures were performed in accordance with the ethical standards of the Helsinki Declaration.

### Study site and participants

The study was conducted within the Legon community in Accra, Ghana, where malaria transmission is limited mainly to the rainy season, typically from March to November. PBMCs from nine participants aged between 25 and 45 years, recruited between November 2013 and March 2014, were used in this study. These participants were from a pool of about 20 subjects who had previously participated in ELISpot standardization studies conducted jointly by NMIMR and the Naval Medical Research Center (NMRC), USA, and their HLA supertype data were available. Eligibility criteria for the current study were as follows: age 18–55 years, normal screening medical history and physical examination, haemoglobin >10 g/dl, absence of known immunodeficiency (> 400 CD4+ T cells/µl of blood) and availability of enough cryo-preserved PBMCs for conduct of assays. On the basis of these, PBMCs from nine HLA typed subjects were available for inclusion in this study. At the time of blood draw, study subjects were screened for malaria parasites by light microscopy.

### Sample collection and processing

Blood sample collection and PBMC isolation were performed as previously described [24,25] and PBMCs were stored in 20 million cells/ml aliquots in liquid nitrogen. For the current study, PBMCs were thawed at 37˚C, washed twice with R5 medium (RPMI-1640 with 5% foetal calf serum, 1% penicillin-streptomycin) and allowed to rest in an incubator at 37˚C, 5% $CO_2$ for up to 8 hours. After this period, PBMCs were again washed and re-suspended in HR10 medium (RPMI 1640 supplemented with 10% normal human serum, 1% penicillin-streptomycin, 1% glutamine) before use in ELISpot assays. For each subject, concentration of the PBMC suspension was adjusted to 4 million/ml before use in assays (final concentration of 400,000 cells/well).

### Negative selection of CD8+ T cell fraction of PBMCs

To confirm that CD8+ T cells responded to peptide stimulants, the CD8+ T cell fraction of PBMCs was enriched by a negative selection protocol that depleted cell types expressing the CD4 receptor, including CD4+ T cells. This was carried out with the anti-human MyOne™ SA Dynabeads® kit (Invitrogen, Life Technologies) and following the bead manufacturer's instructions. In brief, a cocktail of biotinylated mouse anti-human antibodies against non-

CD8+ T cells was added to thawed and washed PBMCs at the required concentration for ELISpot (4 million cells/ml) and incubated for 20 minutes. PBMCs were subsequently incubated with streptavidin-conjugated anti-human MyOne™ SA dynabeads for 15 minutes at 25˚C and CD8+ cells separated from non-CD8+ cells in a magnetic field.

Flow cytometry was used to confirm the effectiveness of the CD8+ T cell enrichment procedure. Three microliters (3 μl) each of mouse anti-human CD4 and CD8 antibodies (BD Pharmingen) were respectively added to 100 μl of whole and CD8+ T cell-enriched PBMCs. The cells were incubated at 4˚C for 20 minutes and washed with 2 ml of R5 medium. Cells were reconstituted in 250 μl of FACs buffer (BD FACSFlow) and acquired (100,000 events) on a BD FACSCalibur machine. Data analysis was performed using the CellQuest™ Pro software (version 6).

The CD8+ T cell-enriched PBMC fraction was then re-suspended in the same starting volume of HR10 medium and used in ELISpot assays (described below) alongside the unfractionated PBMCs.

### Synthetic peptides

Whole and CD8+ T cell-enriched PBMCs were stimulated with 8-10mer single peptides from the 3D7 *P. falciparum* CSP antigen (GenBank accession number X15363). Twenty three 8-10mer CSP peptides were selected either on the basis of being present in peptide pools that gave positive ELISpot responses previously (Cp1, Cp4, Cp6 and Cp9, [23,24]), and/or having been predicted (NetMHC) to be restricted by subject HLA-A or HLA-B allele types (Table 1). The NetMHC algorithm [26] predictions were performed as previously described [24]. Peptides were synthesized by Alpha Diagnostics Inc. (San Antonio, TX, USA) to a purity of > 90%. The locations of the 23 peptides within the previously tested 15mer peptides [23,24] are presented in Table 1. All peptides were diluted to the required concentration with HR10 medium for use in ELISpot assays.

### *Ex vivo* ELISpot IFN-γ assays

IFN-γ ELISpot assays were performed as previously described [24] using cryopreserved PBMCs. PBMCs (unfractionated and the CD8+ T cell-enriched fraction) from subjects were tested in triplicate with 10 μg/ml each of subject-specific CSP peptides. Phytohaemaglutinin (PHA, Sigma Aldrich, USA) (1 μg/ml), concanavalin A (Con A, Sigma Aldrich, USA) (0.625 μg/ml) and a pool of HLA class I-restricted T cell epitopes from common viruses (CEF, Cellular Technology Ltd, USA) (2.0 μg/ml) were used as positive controls and subject PBMCs incubated with culture medium only were used as negative controls. After substrate incubation and plate development, the number of IFN-γ-producing cells was estimated using an automated ELISpot plate reader (AID GmbH, Germany) and the data exported into Microsoft Excel for analysis.

### Data analysis

The mean of replicate readings for each stimulant was calculated and activities were expressed as spot forming cells per million (sfc/m) PBMCs. Any single value for the triplicate readings of each stimulant/control that contributed more than 50% of the standard deviation of the triplicate and was at least three times greater or less than the mean of the remaining two values was considered an outlier and discarded. The assay was considered positive if there was (1) at least a doubling of sfc/m in test wells relative to control wells, and (2) a difference of at least ten spots between test and control wells. This definition was validated and adapted for use in previous studies [24,25].

**Table 1. Peptides used to stimulate study subject PBMCs.**

| Subject | HLA types | Peptides | Amino acids | CSP pool[#] | Subject | HLA types | Peptides | Amino acids | CSP pool[#] |
|---|---|---|---|---|---|---|---|---|---|
| v01 | | ILSVSSFLFV | 7–16 | Cp1 | v05 | A03 | LAILSVSSF | 5–13 | Cp1 |
| | | MPNDPNRNV | 285–293 | Cp5 | | B07/B44 | FVEALFQEY | 15–23 | Cp1 |
| | A02 | HIKEYLNKI | 315–323 | Cp6 | | | RIKPGSANK | 345–353 | Cp7 |
| | B07/B44 | YLNKIQNSL | 319–327 | Cp6 | v06 | | LAILSVSSF | 5–13 | Cp1 |
| | | SVFNVVNSSI | 376–385 | Cp9 | | | SVSSFLFVEA | 9–18 | Cp1 |
| | | GLIMVLSFL | 386–394 | Cp9 | | | FVEALFQEY | 15–23 | Cp1 |
| v02 | | MMRKLAILSV | 1–10 | Cp1 | | A01/A02 | NYDNAGTNLY | 39–48 | Cp2 |
| | | ILSVSSFLFV | 7–16 | Cp1 | | B07 | LYNELEMNYY | 47–56 | Cp2 |
| | | FVEALFQEY | 15–23 | Cp1 | | | SVTCGNGIQV | 335–343 | Cp7 |
| | A01/A02 | NYDNAGTNLY | 39–48 | Cp2 | | | SVFNVVNSSI | 376–385 | Cp9 |
| | B07/B44 | LYNELEMNYY | 47–56 | Cp2 | | | SSIGLIMVL | 383–391 | Cp9 |
| | | YLNKIQNSL | 319–327 | Cp6 | v07 | A03/A24 | LAILSVSSF | 5–13 | Cp1 |
| | | GLIMVLSFL | 386–394 | Cp9 | | B07 | SFLFVEALF | 12–20 | Cp1 |
| v03 | A01/A03 | MPNDPNRNV | 285–293 | Cp5 | v08 | | ALFQEYQCY | 18–26 | Cp1 |
| | B07 | HIKEYLNKI | 315–323 | Cp6 | | | NYDNAGTNLY | 39–48 | Cp2 |
| | | RIKPGSANK | 345–353 | Cp7 | | | LYNELEMNYY | 47–56 | Cp2 |
| v04 | | MMRKLAILSV | 1–10 | Cp1 | | A01/A03 | YLNKIQNSL | 319–327 | Cp6 |
| | | ILSVSSFLFV | 7–16 | Cp1 | | B27 | IQNSLSTEW | 323–331 | Cp6 |
| | | SVSSFLFVEA | 9–18 | Cp1 | | | KMEKCSSVF | 370–378 | Cp9 |
| | | FLFVEALFQE | 13–22 | Cp1 | | | IMVLSFLFL | 388–396 | Cp9 |
| | A02 | YLNKIQNSL | 319–327 | Cp6 | v09 | | LAILSVSSF | 5–13 | Cp1 |
| | | SVTCGNGIQV | 335–343 | Cp7 | | | MPNDPNRNV | 285–293 | Cp5 |
| | | SVFNVVNSSI | 376–385 | Cp9 | | B07/B58 | IQNSLSTEW | 323–331 | Cp6 |
| | | GLIMVLSFL | 386–394 | Cp9 | | | SSIGLIMVL | 383–391 | Cp9 |
| | | | | | | | LIMVLSFLF | 387–395 | Cp9 |

Peripheral blood mononuclear cells (PBMCs) from were stimulated with the 8-10mer peptides indicated for each subject. Peptides were predicted to bind to the subjects' indicated HLA types using the artificial neural network-based NetMHC algorithm.

# Indicates the CSP 15mer overlapping peptide pool, as described by Ganeshan et al.[24] that contains the predicted -10mer peptide

## Results

### IFN-γ responses in whole PBMC fractions

All subjects used in this study had a normal medical history at screening and were negative for malaria by light microscopy. Cryopreserved PBMCs from the nine subjects were retrieved and used in this study. For analysis and comparison, IFN-γ ELISpot response (sfc/m) for the unstimulated PBMC control are subtracted from the responses for each test peptide. All subjects responded positively to the mitogens Con A (response range 75–379 sfc/m, after unstimulated background subtraction) and PHA (122–467 sfc/m) but as expected, responses of individual subjects to CEF were variable, ranging from no response (v08, 0 sfc/m), a low response (v09, 25 sfc/m) to high responses (v01 to v07, 233–513 sfc/m). In all assays, unstimulated medium control was low and responses ranged between 1 and 24 sfc/m PBMCs.

A total of 23 peptides were used to stimulate PBMCs from the nine study subjects. The least number of peptides tested per subject was two for subject v07 whilst the highest number tested per subject was eight for each of subjects v04, v06 and v08 (Table 2). In all, five subjects responded to a total of ten peptides: subject v03 responded to three peptides (MPNDPNRNV, HIKEYLNKI and RIKPGSANK), subject v04 responded to three peptides (SVSSFLFVEA,

FLFVEALFQE, GLIMVLSFL), subject v07 responded to two peptides (LAILSVSSF and SFLFVEALF), subject v08 responded to one peptide (YLNKIQNSL) and subject v09 responded to two peptides (LAILSVSSF and IQNSLSTEW) (Table 2). Thus two subjects (v04 and v09) made responses to peptide LAILSVSSF. Of the ten positive responses, subject v07's response to peptide SFLFVEALF recalled the highest activity (38 sfc/m PBMCs), while subjects v04 and v08 had the lowest activities (10 sfc/m PBMCs) after unstimulated background subtraction. IFN-γ responses of the four remaining subjects did not meet our positivity definition criteria.

## IFN-γ responses in CD8+ T cell-enriched PBMCs

There were sufficient cryopreserved PBMC from eight of the nine study subjects for CD8+ T cell enrichment studies. CD8+ T cell enrichment was assessed by surface staining for CD4 and

**Table 2. Interferon gamma responses by unfractionated PBMCs to the 8-10mer CSP peptide peptides.**

| Subject | HLA types | Stimulants | sfc/m | Response | Subject | HLA types | Stimulants | sfc/m | Response |
|---------|-----------|------------|-------|----------|---------|-----------|------------|-------|----------|
| v01 | | ILSVSSFLFV | 27 | neg | v05 | A03 | LAILSVSSF | 24 | neg |
| | | MPNDPNRNV | 18 | neg | | B07/B44 | FVEALFQEY | 24 | neg |
| | A02 | HIKEYLNKI | 26 | neg | | | RIKPGSANK | 13 | neg |
| | B07/B44 | YLNKIQNSL | 23 | neg | | | Medium | 10 | |
| | | SVFNVVNSSI | 22 | neg | v06 | | LAILSVSSF | 6 | neg |
| | | GLIMVLSFL | 25 | neg | | | SVSSFLFVEA | 6 | neg |
| | | Medium | 17 | | | A01/A02 | FVEALFQEY | 7 | neg |
| v02 | | MMRKLAILSV | 12 | neg | | B07 | NYDNAGTNLY | 6 | neg |
| | A01/A02 | ILSVSSFLFV | 8 | neg | | | LYNELEMNYY | 6 | neg |
| | B07/B44 | FVEALFQEY | 7 | neg | | | SVTCGNGIQV | 6 | neg |
| | | NYDNAGTNLY | 5 | neg | | | SVFNVVNSSI | 6 | neg |
| | | LYNELEMNYY | 13 | neg | | | SSIGLIMVL | 6 | neg |
| | | YLNKIQNSL | 11 | neg | | | Medium | 6 | |
| | | GLIMVLSFL | 7 | neg | v07 | A03/A24 | **LAILSVSSF** | **33** | **POS** |
| | | Medium | 4 | | | B07 | **SFLFVEALF** | **53** | **POS** |
| v03 | A01/A03 | **MPNDPNRNV** | **23** | **POS** | | | Medium | 15 | |
| | B07 | **HIKEYLNKI** | **26** | **POS** | v08 | | LAILSVSSF | 3 | neg |
| | | **RIKPGSANK** | **31** | **POS** | | | ALFQEYQCY | 5 | neg |
| | | Medium | 9 | | | | NYDNAGTNLY | 3 | neg |
| v04 | | MMRKLAILSV | 8 | neg | | A01/A03 | LYNELEMNYY | 3 | neg |
| | | ILSVSSFLFV | 13 | neg | | B27 | **YLNKIQNSL** | **13** | **POS** |
| | | **SVSSFLFVEA** | **18** | **POS** | | | IQNSLSTEW | 3 | neg |
| | A02 | **FLFVEALFQE** | **23** | **POS** | | | KMEKCSSVF | 5 | neg |
| | | YLNKIQNSL | 13 | neg | | | IMVLSFLFL | 6 | neg |
| | | SVTCGNGIQV | 11 | neg | | | Medium | 3 | |
| | | SVFNVVNSSI | 12 | neg | v09 | | **LAILSVSSF** | **50** | **POS** |
| | | **GLIMVLSFL** | **33** | **POS** | | A01/A03 | MPNDPNRNV | 45 | neg |
| | | Medium | 8 | | | B07/B58 | **IQNSLSTEW** | **48** | **POS** |
| | | | | | | | SSIGLIMVL | 39 | neg |
| | | | | | | | LIMVLSFLF | 41 | neg |
| | | | | | | | Medium | 24 | |

Peptides were used to stimulate subject PBMCs (400,000 cells/well) in triplicate, and the number of cells that were actively secreting peptide-specific IFN-γ enumerated and expressed as the average number of spot forming cells per million (sfc/m) PBMCs. The presented sfc/m data are the absolute counts and the medium/background responses. Response positivity criteria have been described under the "Methods" section. POS = a positive peptide response, also indicated in bold, neg = negative peptide response.

**Table 3. Interferon-gamma responses by CD8+ PBMC fractions to the 8-10mer CSP peptides.**

| Subject | HLA types | Stimulants | sfc/m | Response | Subject | HLA types | Stimulants | sfc/m | Response |
|---------|-----------|------------|-------|----------|---------|-----------|------------|-------|----------|
| v02 |  | MMRKLAILSV | 10 | neg | v06 |  | LAILSVSSF | 1 | neg |
|  |  | ILSVSSFLFV | 9 | neg |  |  | SVSSFLFVEA | 1 | neg |
|  | A01/A02 | FVEALFQEY | 6 | neg |  |  | FVEALFQEY | 4 | neg |
|  | B07/B44 | NYDNAGTNLY | 10 | neg |  | A01/A02 | NYDNAGTNLY | 2 | neg |
|  |  | LYNELEMNYY | 6 | neg |  | B07 | LYNELEMNYY | 2 | neg |
|  |  | YLNKIQNSL | 12 | neg |  |  | SVTCGNGIQV | 1 | neg |
|  |  | GLIMVLSFL | 6 | neg |  |  | SVFNVVNSSI | 1 | neg |
|  |  | Medium | 6 |  |  |  | SSIGLIMVL | 2 | neg |
| v03 |  | **MPNDPNRNV** | **36** | **POS** |  |  | Medium | 1 |  |
|  | A01/A03 | HIKEYLNKI | 28 | neg | v07 | A03/A24 | LAILSVSSF | 15 | neg |
|  | B07 | **RIKPGSANK** | **39** | **POS** |  | B07 | SFLFVEALF | 21 | neg |
|  |  | Medium | 16 |  |  |  | Medium | 12 |  |
| v04 |  | MMRKLAILSV | 8 | neg | v08 |  | LAILSVSSF | 4 | neg |
|  |  | ILSVSSFLFV | 6 | neg |  |  | ALFQEYQCY | 3 | neg |
|  |  | SVSSFLFVEA | 12 | neg |  |  | NYDNAGTNLY | 3 | neg |
|  |  | FLFVEALFQE | 12 | neg |  | A01/A03 | LYNELEMNYY | 3 | neg |
|  | A02 | YLNKIQNSL | 7 | neg |  | B27 | YLNKIQNSL | 3 | neg |
|  |  | SVTCGNGIQV | 10 | neg |  |  | IQNSLSTEW | 4 | neg |
|  |  | **SVFNVVNSSI** | **16** | **POS** |  |  | KMEKCSSVF | 4 | neg |
|  |  | **GLIMVLSFL** | **43** | **POS** |  |  | IMVLSFLFL | 3 | neg |
|  |  | Medium | 5 |  |  |  | Medium | 3 |  |
| v05 |  | LAILSVSSF | 11 | neg | v09 |  | LAILSVSSF | 21 | neg |
|  | A03 | FVEALFQEY | 11 | neg |  | A01/A03 | MPNDPNRNV | 21 | neg |
|  | B07/B44 | **RIKPGSANK** | **38** | **POS** |  | B07/B58 | IQNSLSTEW | 21 | neg |
|  |  | Medium | 11 |  |  |  | SSIGLIMVL | 21 | neg |
|  |  |  |  |  |  |  | LIMVLSFLF | 26 | neg |
|  |  |  |  |  |  |  | Medium | 21 |  |

Peptides were used to stimulate the CD8+ fraction of PBMCs in triplicate, and the number of cells that were actively secreting peptide-specific IFN-γ enumerated and expressed as the average number of spot forming cells per million (sfc/m) PBMCs. The presented sfc/m data are the absolute counts and the medium/background responses. response for specific subjects. POS = a positive peptide response, also indicated in bold, neg = negative peptide response.

CD8 receptors and flow cytometry. The proportion of CD4+ T cells reduced from 42.1 ± 8.8% in the unfractionated population to 1.3 ± 1.1% after depletion, while that of CD8+ T cells increased from 28.4 ± 6.4% in the unfractionated population to 66.2 ± 11.1% after enrichment. Representative data for one of the study subjects is presented in S1 Fig.

In assays with CD8+ enriched cells, five of the nine subjects responded to Con A (response range 16–213 sfc/m after medium/background subtraction), six responded to PHA (57–372 sfc/m) and 5 responded to CEF (26–311 sfc/m). All subjects responded to at least two of the three positive control stimulants, with one subject each responding to only one of the three stimulants. Unstimulated medium control responses for assays with CD8+ enriched cells ranged between 1 and 21 sfc/m PBMCs.

Of the ten peptides that yielded positive ELISpot responses in unfractionated subject PBMCs, three peptides remained positive with the CD8+ T cell-enriched PBMC fraction from the same subjects (Table 3); subject v03 remained positive to two of the three peptides that were previously positive (MPNDPNRNV and RIKPGSANK) while subject v04 remained positive to peptide GLIMVLSFL. In addition, two peptides that were previously negative with

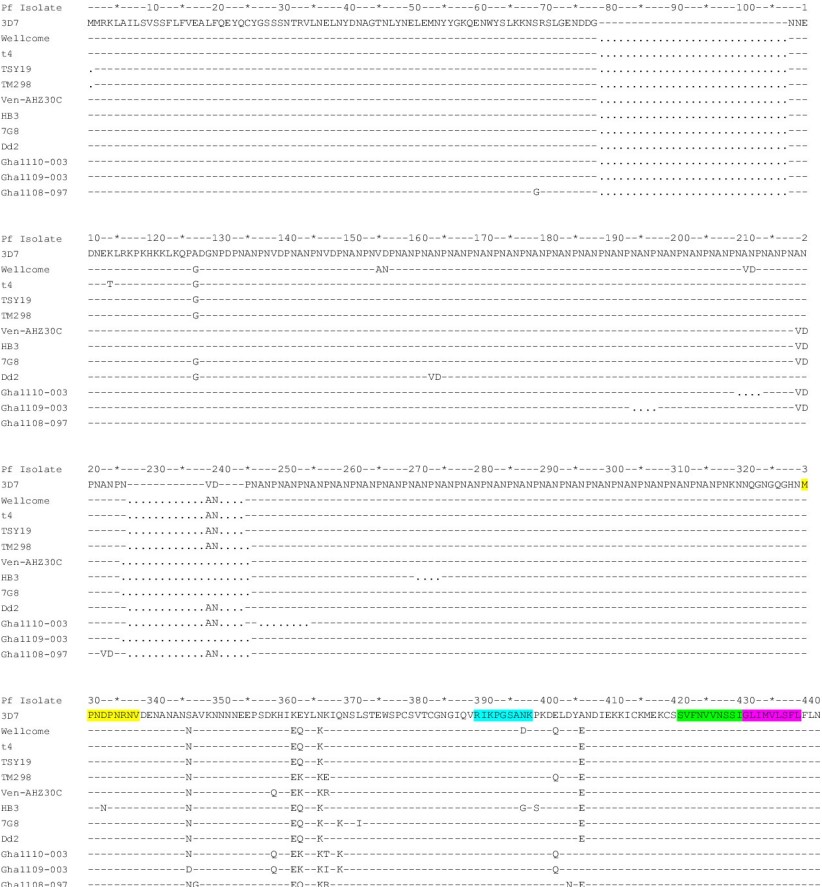

**Fig 1. Alignment of PfCSP sequences from multiple Pf strains.** Sequences include those from established laboratory parasite strains as well field isolates from Ghana, Nigeria, Venezuela, Thailand. Dash lines represent the conserved residues relative to the 3D7 consensus from which tested peptides were derived. Dots represent deleted sequences compared to the Wellcome sequence, which is the longest in terms of amino acid residues (442). The final four peptides identified as being immunodominant in this study are highlighted in different colours. Sequences were sourced from Genbank and UniProt sequence databases, and alignments to identify deleted residues in some sequences were done in UniProt.

unfractionated PBMCs gave positive responses with the CD8-enriched PBMC fractions; v04 tested positive against peptide SVFNVVNSSI and v05 tested positive against peptide RIKPGSANK. Thus a total of four peptides tested positive against the CD8+ T cell-enriched fraction of PBMCs, and these peptides are most likely presented by HLA class I molecules to CD8+ T cells. The four positive peptides are from the C-terminal of the protein and show limited polymorphism (Fig 1). By extension, the loss of positive peptide responses against CD8+ T cell-enriched PBMC fractions from some subjects (a total of seven peptides for subjects v03, v04, v07, v08 and v09) suggests that those peptides are likely to be presented by HLA class II molecules to CD4+ T cells, or that they did not meet the criteria for response positivity.

## Discussion

An effective malaria vaccine is essential to the malaria eradication agenda and there is an urgent need to develop cost-effective broad coverage vaccines. A potentially effective and relatively cheaper approach is to identify HLA-restricted immunodominant epitopes from multiple parasite antigens and incorporate these into subunit multi-epitope vaccines. Such epitopes

may be identified through bioinformatics prediction of peptide binding by HLA using tools such as NetMHC [26], even though not all HLA-bound peptides may be recognized by T cells [27]. It is therefore important to experimentally confirm T cell recognition of, and activation by HLA-bound peptides using T cell-based assays such as ELISpot. Using this approach, we have previously identified parasite antigen-specific 15mer peptide pools that elicit potent IFN-γ responses when tested against PBMCs from malaria exposed individuals [23–25]. It is therefore imperative to further determine the specific peptides within these parasite antigen pools that mediate the induction of these potential protection-associated responses. The aim of this study was therefore to experimentally assess the induction of IFN-γ responses by selected 8-10mer single peptides from *P. falciparum* CSP using PBMCs from HLA-typed subjects with natural exposure to malaria.

IFN-γ responses (sfc/m) measured in this study were generally of the same order of magnitude as those elicited against CSP peptide pools in individuals from the same naturally exposed population [24] but lower than responses achieved in malaria-naïve individuals who have been immunized with CSP-based DNA vaccines [8,14].

Ten of the 23 peptides (43.5%) elicited positive IFN-γ responses in PBMCs from five of the nine HLA-matched study subjects (Table 2), suggesting that the specific HLA alleles expressed by these subjects recognized and presented peptides to T cells. In addition to being predicted as HLA class I-restricted epitopes, six of the ten positive peptides (LAILSVSSF, SFLFVEALF, SVSSFLFVEA, FLFVEALFQE, GLIMVLSFL, RIKPGSANK) also tested positive experimentally in IFN-γ ELISpot assays (identified minimal epitopes underlined) with PBMCs from naïve subjects who have been immunized with DNA or peptide-based malaria vaccines [14]. GLIMVLSFL also tested positive experimentally against PBMCs from individuals naturally exposed to malaria [28]. The current data generated with PBMCs from naturally exposed individuals therefore confirms the immunodominant nature of these peptides, even against PBMCs from individuals living in a very low malaria transmission area. The other four identified positive peptides were YLNKIQNSL, IQNSLSTEW, MPNDPNRNV and HIKEYLNKI. These peptides are all present in a PfCSP long synthetic peptide (aa282–383) vaccine candidate that elicited significant IFN-γ responses in PBMCs from malaria-naïve subjects immunized with this vaccine [29]. Confirmation of previously identified immunodominant epitopes and the experimental identification of additional epitopes collectively give relevance to epitope discovery efforts in malaria endemic populations.

Five of the ten positive peptides are located at the N terminal (LAILSVSSF, SVSSFLFVEA, SFLFVEALF and FLFVEALFQE, collectively within amino acids 1–30) and C-terminal (GLIMVLSFL, amino acids 386–394) ends of the CSP antigen. The peptide MPNDPNRNV (amino acids 285–293) occurs in the central repeat region of the antigen while the other four positive peptides (IQNSLSTEW, HIKEYLNKI, YLNKIQNSL and RIKPGSANK) all occur in the middle of the CSP antigen, outside the repetitive region. All ten positive peptides have very limited or no polymorphic residues, but the extent of sequence conservation of these regions of the CSP antigen and their effect on T cell response induction will need to be further investigated.

To confirm the role of CD8+ T cells, PBMCs were enriched for CD8+ T cells by depletion of CD4+ T cells and the efficiency of depletion confirmed using flow cytometry. Four of the ten peptides (17.4%) showed HLA class I restriction following testing against the CD8-enriched PBMC fraction (Table 3). Three of the four peptides (MPNDPNRNV and RIKPGSANK in subject v03, and GLIMVLSFL in subject v04) were amongst the ten previously positive peptides. RIKPGSANK also tested positive against the CD8-enriched PBMC fraction from subject v05, but not the unfractionated PBMCs from the same subject. This was also the case for the fourth peptide (SVFNVVNSSI); it tested positive with the CD8+ T cell-enriched PBMC

fraction but not the unfractionated PBMCs from subject v04. The reason for this observation is unclear, but it is possible that CD8+ T cells in the unfractionated PBMCs from these subjects were inhibited by CD4+ T cells of the regulatory phenotype present in the PBMCs. There is evidence of CD4+ T cell-mediated inhibition of effector T cell responses in *ex vivo* cultures [30]. Additionally, reduced immune activity and a consequential early increase in parasite burden in individuals with high levels of CD4+ Treg cells has been demonstrated [31]. The observation however may also simply be a result of the responses against the unfractionated PBMCs not meeting the positivity criteria. Although the CD4 depletion kit used in this study may also have depleted some subsets of antigen presenting cells (APCs), the observed response of four out of the ten positive peptides with unfractionated PBMCs suggests that there was effective antigen presentation. This limitation in our approach however indicates that with the full complement of APCs, the CD8+ T cell response to the positive peptides could have been greater in magnitude than was measured. It is also possible that with the full complement of APCs, additional positive peptides. This also reflects in responses to the CEF and Con A positive controls, which are generally lower in the CD8 enriched fractions compared to unfractionated PBMCs.

Two of the four peptides identified as being HLA class I-restricted (GLIMVLSFL and SVFNVVNSSI) are present in the previously positive peptide pool Cp9 at the C-terminal end of the CSP antigen [24]. The two peptides have been predicted by NetMHC to be HLA A02-restricted, and this has been experimentally confirmed in assays with CD8+ T cells from a naturally exposed individual (subject v04) in this study as well as from an immunized malaria-naïve individual [14], both of whom express the HLA A02 phenotype. The HLA A02 restriction of GLIMVLSFL and its recognition and presentation by other HLA class I supertypes has been demonstrated in sporozoite-immunized as well as naturally exposed subjects [28,32]. Peptide SVFNVVNSSI, aside the observation that it is HLA A02-restricted, has also been predicted to bind to HLA A24 and B27 supertypes in naturally exposed subjects [24] and to HLA A01 supertypes in vaccinated malaria naïve subjects [14].

The other two peptides that were positive against CD8+ T cell-enriched PBMC fractions have been predicted to be restricted by HLA B07 (MPNDPNRNV) and HLA A01A03 and HLA A03 (RIKPGSANK), respectively. Subject v03 who responded to both of these peptides correspondingly expresses both the HLA A03 and HLA B07 phenotypes (Table 3). MPNDPNRNV has previously been identified as an epitope in subjects from naturally exposed subjects [33,34] and is present in the CSP peptide pool Cp5 [24]. RIKPGSANK has been described as an HLA A03-restricted class I epitope that is part of a CSP long synthetic peptide vaccine candidate that elicited potent T cell responses in malaria naïve adults [29]; it is also present in peptide pools Cp7/Cp8 [24], though none of these two peptide pools have tested positive in our previous assays with PBMCs from malaria exposed individuals. It is however possible that these peptides were presented by promiscuous HLA allele types expressed by subject v03. HLA binding promiscuity is a well-known phenomenon [28,35,36] and that makes the experimental assessment of T cell activation by peptides, beyond bioinformatics predictions, very important. Most of the HLA supertypes that recognize and present the four identified peptides are believed to occur at high frequency in many ethnicities including Africans [27,37], hence HLA restriction of T cell responses to these peptides may ultimately not be the major obstacle to this T cell-based vaccine development approach.

The seven peptides that tested positive with unfractionated PBMCs but negative with the CD8+ T cell-enriched PBMC fraction have also been predicted by NetMHC to be HLA class I restricted; LAILSVSSF (in peptide pool Cp1) is predicted to be HLA B07-, B27- and B58-restricted, SFLFVEALF (in pool Cp1) is HLA A24-retricted, SVSSFLFVEA and FLFVEALFQE (both in pool Cp1) are HLA A02-restricted, IQNSLSTEW (in pools Cp6, Cp7) is HLA B27- and HLA B58-restricted, HIKEYLNKI (in pool Cp6) is HLA A02- and HLA A01A03-restricted and

YLNKIQNSL (in pool Cp6) is HLA A02- and B27-restricted [24]. As earlier stated, these peptides did not elicit positive IFN-γ responses in CD8-enriched PBMC fractions, even though the respective subjects generally express the expected HLA supertypes. This experimental outcome suggests that these peptides may have activated CD4+ T cells in the unfractionated PBMCs. A number of studies have reported a high degree of overlap between HLA class I- and class II-restricted epitopes in *P. falciparum* [38] as well as other pathogen [39,40] antigens. This might explain why peptides predicted to be HLA class I-restricted would elicit what appears to be HLA class II-dependent IFN-γ responses. Interestingly, six of the seven peptides (LAILSVSSF, SVSSFLFVEA, FLFVEALFQE, IQNSLSTEW, HIKEYLNKI and RIKPGSANK) which were positive against unfractionated PBMCs, but not the CD8+ T cell-enriched PBMC fractions, have previously been confirmed as parts of the sequences of class II-restricted epitopes [38,41,42]. It is also possible that the lack of HLA class I-specific positive responses against these peptides could be due to the fact that the specific HLA class I alleles expressed by our study subjects are different from those that are capable of optimally binding and presenting these peptides to CD8+ T cells.

Some limitations of the current study include our inability to assess the immune status of subjects at the time of drawing blood samples as well as to identify potential infections by PCR as these would have aided the interpretation of study data. Also, we were unable to correspondingly deplete CD8+ T cells and test peptides against CD4+ T cell enriched PBMC fractions due to limited availability of cryopreserved samples. Additionally, it is possible that the CD4+ cell depletion procedure we employed might have depleted some antigen presenting cell subsets, and this could limit the magnitude of responses and hence the number of positive peptides observed. We will in the future employ intracellular cytokine staining methods for performing immune cell subset analysis. These notwithstanding, the data presented make an important contribution to the search for essential parasite targets for vaccine development purposes.

In summary, the study has identified four immunodominant HLA class I-restricted epitopes within the *P. falciparum* CSP antigen. All four peptides have been previously reported as epitopes on the basis of experimental data with PBMCs from either vaccinated malaria-naïve subjects or naturally exposed subjects. These epitopes show limited or no polymorphism with the possibility of being recognized and presented to CD8+ T cells by multiple HLA supertypes. A demonstration of anti-malarial protection in naturally exposed individuals as a result of potent T cell responses to these peptides remains to be established. If such protective role is confirmed, these peptides will be important candidates for inclusion in subunit, multi-epitope, strain-transcending, T cell-based malaria vaccines that are capable of inducing IFN-γ responses in endemic populations with diverse genetic backgrounds. Our data also reinforces the need to undertake epitope identification studies in naturally exposed individuals as part of malaria vaccine design strategies.

## Supporting information

**S1 Fig. Representative histograms of T cell populations before and after CD4+ cell depletion.** PBMCs from eight of the nine study subjects were depleted of cells expressing the CD4 receptor for the purpose of assessing the T cell lineage of peptide-specific IFN- responses. PBMC from all such depletions showed very similar results. Proportions indicate the percentage of total cells gated.
(TIF)

## Acknowledgments

We thank all the subjects who participated in this study.

## Copyright statement

M.S. and E.V. are employees of the U.S. Government. This work was prepared as part of their official duties. Title 17 U.S.C. 105 provides that "Copyright protection under this title is not available for any work of the United States Government." Title 17 U.S.C. 101 defines a U.S. Government work as work prepared by a military service member or employee of the U.S. Government as part of that person's official duties

## Author Contributions

**Conceptualization:** Kwadwo A. Kusi, Linda E. Amoah, Michael Hollingdale, Bjoern Peters, Daniel Dodoo, Eileen Villasante, Martha Sedegah.

**Data curation:** Kwadwo A. Kusi, Dorothy Anum, Yvonne Nartey, Harini Ganeshan, Maria Belmonte, Yohan Kim, John Tetteh, Eric Kyei-Baafour, Martha Sedegah.

**Formal analysis:** Kwadwo A. Kusi, Felix E. Aggor, Yvonne Nartey, Michael Hollingdale, Eric Kyei-Baafour.

**Funding acquisition:** Kwadwo A. Kusi, Linda E. Amoah.

**Investigation:** Kwadwo A. Kusi, Felix E. Aggor, Yvonne Nartey, Daniel Amoako-Sakyi, Dorcas Obiri-Yeboah, Michael Hollingdale, Harini Ganeshan, Maria Belmonte, Bjoern Peters, Yohan Kim, John Tetteh, Eric Kyei-Baafour, Daniel Dodoo, Martha Sedegah.

**Methodology:** Kwadwo A. Kusi, Felix E. Aggor, Dorothy Anum, Yvonne Nartey, Eric Kyei-Baafour.

**Project administration:** Kwadwo A. Kusi.

**Resources:** Kwadwo A. Kusi, Linda E. Amoah, Michael Hollingdale, Maria Belmonte, Bjoern Peters, Yohan Kim, John Tetteh, Daniel Dodoo, Eileen Villasante, Martha Sedegah.

**Software:** Kwadwo A. Kusi, Bjoern Peters, Yohan Kim.

**Supervision:** Kwadwo A. Kusi, Daniel Amoako-Sakyi, Dorcas Obiri-Yeboah.

**Visualization:** Kwadwo A. Kusi, John Tetteh, Martha Sedegah.

**Writing – original draft:** Kwadwo A. Kusi, Felix E. Aggor.

**Writing – review & editing:** Linda E. Amoah, Dorothy Anum, Yvonne Nartey, Daniel Amoako-Sakyi, Dorcas Obiri-Yeboah, Michael Hollingdale, Harini Ganeshan, Maria Belmonte, Bjoern Peters, Yohan Kim, John Tetteh, Eric Kyei-Baafour, Daniel Dodoo, Eileen Villasante, Martha Sedegah.

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
