## [Decision Letter · Decision Letter 0]

22 Oct 2019

PONE-D-19-25608

Identification of Plasmodium falciparum circumsporozoite protein-specific CD8+ T cell epitopes in a malaria exposed population

PLOS ONE

Dear Dr Kusi,

Thank you for submitting your manuscript to PLOS ONE. After careful consideration, we feel that it has merit but does not fully meet PLOS ONE’s publication criteria as it currently stands. Therefore, we invite you to submit a revised version of the manuscript that addresses the points raised during the review process.

I fully agree with the expert reviewer's comments appended. Any revision must address the concerns raised with respect to assessment of CD8 T cell responses and the the method used for depletion of cells in that regard, and should modify the conclusions drawn appropriately.

We would appreciate receiving your revised manuscript by Dec 06 2019 11:59PM. To enhance the reproducibility of your results, we recommend that if applicable you deposit your laboratory protocols in protocols.io, where a protocol can be assigned its own identifier (DOI) such that it can be cited independently in the future. For instructions see: http://journals.plos.org/plosone/s/submission-guidelines#loc-laboratory-protocols

We look forward to receiving your revised manuscript.

Kind regards,

Adrian J.F. Luty, PhD

Academic Editor

PLOS ONE

**Journal Requirements:**

**Comments to the Author**

1. Is the manuscript technically sound, and do the data support the conclusions?

Reviewer #1: Partly

2. Has the statistical analysis been performed appropriately and rigorously? 

Reviewer #1: Yes

3. Have the authors made all data underlying the findings in their manuscript fully available?

Reviewer #1: Yes

4. Is the manuscript presented in an intelligible fashion and written in standard English?

Reviewer #1: Yes

5. Review Comments to the Author

Reviewer #1: This manuscript addresses the fine mapping of CD8 T cell epitopes on the P. falciparum Circumsporozoite Protein recognised by PBMCs from naturally-exposed Ghanaian donors. This forms a continuation of previous work by the authors in which they already demonstrated recognition of larger peptide pools spanning these epitopes by Ghanaian donors; the majority of these epitopes are also recognised by sporozoite vaccinees. Experimental validation of epitope prediction algorithms is crucial prior to such epitopes being selected for use in a multi-epitope sub-unit pre-erythrocytic malaria vaccine, which may form an alternative to attenuated whole-sporozoite approaches.

As proof of principle, the authors identify several such CD8-restricted epitopes, which moreover appear to be conserved across P.f. strains and may be recognised by multiple HLA supertypes (both of which are advantageous for inclusion in a vaccine). Unfortunately, fewer predicted peptides than perhaps expected could be conclusively shown to be recognised (particularly in the 'CD8-enriched' cultures, see also below). It is also perhaps slightly concerning that the 'HLA-promiscuous' epitopes are not recognised more broadly. Finally, as the authors acknowledge, association of any of these epitopes with protection remains to de demonstrated.

The study approach is generally clear and appropriate. The manuscript is well-written and the abstract and discussion balanced, including addressing unexpected results and most limitations. Ethical approval is in place.

The immunological methodology does suffer from some limitations, which presumably can no longer be addressed experimentally, but could perhaps be addressed a little more extensively in the discussion.

Although highly suggestive, the depletion of CD4+ T cells does not conclusively prove that the remaining remaining IFNg must be due to CD8+ T cells. Did the authors not consider depleting CD8-expressing cells in first instance instead of CD4-expressing cells, in order to directly demonstrate CD8+ T cells' role? An obvious obvious alternative route would have been flow cytometry, but presumably this was not available?

What kit exactly was used for negative selection of CD8+ T cells? The M&M section mentions variously '[depleting] all cell types expressing the CD4 receptor' and 'a cocktail of ... antibodies against non-CD8+ T cells'. Depending on this, DCs, monocytes (both of which may also express CD4) and/or B-cells may have been depleted from the PBMCs in addition to CD4+ T cells, limiting the availability of APCs to (cross-)present the peptide to the remaining CD8+ T cells. May this partly explain the lower than expected sfc count in the 'CD8-enriched' samples compared to whole PBMC samples for many of the peptides (which were all supposed to be MHC-I restricted)? Were responses to CEF also lower in the 'CD8-enriched' samples?

The authors apply response positivity criteria used and validated in their previous studies. Nevertheless, given (perhaps not unexpectedly) the generally marginal responses to individual peptides (corrected 0-38 sfc/m) in comparison to the range of the neg control (1-24 sfc/m), could the authors in table 2 and 3 maybe provide for each subject the actual sfc value of the neg control for respectively whole PBMC and CD8-enriched cultures? Responses to individual peptides could then be shown either as absolute counts or, as currently, corrected counts. Either way, the reader will be able to form a slightly better impression of the relative strength of individual peptide responses. Were neg control responses generally also lower in the 'CD8-enriched' samples than the unfractionated samples? May this explain why 'positivity criteria' for e.g. SVFNVVNSSI were not met in the unfractionated PBMCs?

6. PLOS authors have the option to publish the peer review history of their article (what does this mean?). If published, this will include your full peer review and any attached files.

Reviewer #1: Yes: Matthew B.B. McCall

---

## [Author Response · Author response to Decision Letter 0]

2 Dec 2019

Responses to reviewer comments

We thank the reviewer for the very insightful comments, and have provided a point-by-point response to these below;

Reviewer #1: This manuscript addresses the fine mapping of CD8 T cell epitopes on the P. falciparum Circumsporozoite Protein recognised by PBMCs from naturally-exposed Ghanaian donors. This forms a continuation of previous work by the authors in which they already demonstrated recognition of larger peptide pools spanning these epitopes by Ghanaian donors; the majority of these epitopes are also recognised by sporozoite vaccinees. Experimental validation of epitope prediction algorithms is crucial prior to such epitopes being selected for use in a multi-epitope sub-unit pre-erythrocytic malaria vaccine, which may form an alternative to attenuated whole-sporozoite approaches.

Reviewer comment

As proof of principle, the authors identify several such CD8-restricted epitopes, which moreover appear to be conserved across P.f. strains and may be recognised by multiple HLA supertypes (both of which are advantageous for inclusion in a vaccine). Unfortunately, fewer predicted peptides than perhaps expected could be conclusively shown to be recognised (particularly in the 'CD8-enriched' cultures, see also below). It is also perhaps slightly concerning that the 'HLA-promiscuous' epitopes are not recognised more broadly. Finally, as the authors acknowledge, association of any of these epitopes with protection remains to be demonstrated.

Response

Only a very limited number of peptides (between 3 and 8) were tested against PBMCs from each of the 9 study volunteers (due to limited cell numbers). We therefore did not expect to identify many epitopes per individual. What we sought to show is how many of the positive peptides with whole PBMCs are HLA class I-restricted, since there is also the possibility of having some of the immunodominant epitopes to be HLA class II-restricted. 

The HLA diversity in these 9 volunteers may be too limited to actually demonstrate binding promiscuity. It is also important to note that promiscuity in HLA binding as described in the manuscript is mostly based on predictive algorithms which have been duly referenced and not necessarily on experimental data. 

Regarding the statement on association with protection, we used PBMCs from adult volunteers with a history of natural exposure to malaria infections, but we did not have any information on their antimalarial protection status, hence that statement. We are currently working on addressing this, in a study where we have recruited people with a history of recent exposure to infectious bites and sometimes even show malaria parasites in the blood (an indication of a completed liver stage cycle) but with no clinical symptoms of malaria. Thus in our current on-going study, we have defined “protection” as evidence of exposure to recent infectious bites that do not result in clinical malaria symptoms.

Reviewer comment

The study approach is generally clear and appropriate. The manuscript is well-written and the abstract and discussion balanced, including addressing unexpected results and most limitations. Ethical approval is in place.

The immunological methodology does suffer from some limitations, which presumably can no longer be addressed experimentally, but could perhaps be addressed a little more extensively in the discussion.

Response: We have included additional study limitations and how they impact the presented data to the discussion section. 

Although highly suggestive, the depletion of CD4+ T cells does not conclusively prove that the remaining IFNg must be due to CD8+ T cells. Did the authors not consider depleting CD8-expressing cells in first instance instead of CD4-expressing cells, in order to directly demonstrate CD8+ T cells' role? An obvious alternative route would have been flow cytometry, but presumably this was not available?

Response

We initially considered testing both CD8-enriched and CD4 enriched T cells for comparison with whole PBMCs, but had to go with just one of them because of the limited cell numbers

Reviewer comment

What kit exactly was used for negative selection of CD8+ T cells? The M&M section mentions variously '[depleting] all cell types expressing the CD4 receptor' and 'a cocktail of ... antibodies against non-CD8+ T cells'. Depending on this, DCs, monocytes (both of which may also express CD4) and/or B-cells may have been depleted from the PBMCs in addition to CD4+ T cells, limiting the availability of APCs to (cross-)present the peptide to the remaining CD8+ T cells. May this partly explain the lower than expected sfc count in the 'CD8-enriched' samples compared to whole PBMC samples for many of the peptides (which were all supposed to be MHC-I restricted)? Were responses to CEF also lower in the 'CD8-enriched' samples?

Response

We used the anti-human MyOne^™^ SA Dynabeads® kit (Invitrogen, Life Technologies) for CD8 T cell enrichment, and we agree with the reviewer regarding depletion of some antigen presenting cells (APCs) along with CD4 T cells since some APC subsets may also express the targeted receptors. It is however important to note that some other APC subsets likely to remain to undertake antigen cross-presentation, although we agree with the reviewer that these might have lowered the observed epitope positivity rate. This would then mean that our responses could have been stronger than observed, in the presence of the full complement of APCs.

All volunteers responded positively to the at least of the three positive controls

Reviewer comment

The authors apply response positivity criteria used and validated in their previous studies. Nevertheless, given (perhaps not unexpectedly) the generally marginal responses to individual peptides (corrected 0-38 sfc/m) in comparison to the range of the neg control (1-24 sfc/m), could the authors in table 2 and 3 maybe provide for each subject the actual sfc value of the neg control for respectively whole PBMC and CD8-enriched cultures? Responses to individual peptides could then be shown either as absolute counts or, as currently, corrected counts. Either way, the reader will be able to form a slightly better impression of the relative strength of individual peptide responses. Were neg control responses generally also lower in the 'CD8-enriched' samples than the unfractionated samples? May this explain why 'positivity criteria' for e.g. SVFNVVNSSI were not met in the unfractionated PBMCs?

Response

We agree with the reviewer and have provided the respective medium/background control values in Tables 2 and 3. We have therefore reverted to absolute counts (without background subtraction) for each stimulant in these two tables. We have also incorporated data on the positive control and background counts for assays with CD8+ T cell enriched PBMCs in the results section.

---

## [Decision Letter · Decision Letter 1]

9 Jan 2020

Identification of Plasmodium falciparum circumsporozoite protein-specific CD8+ T cell epitopes in a malaria exposed population

PONE-D-19-25608R1

Dear Dr. Kusi,

We are pleased to inform you that your manuscript has been judged scientifically suitable for publication and will be formally accepted for publication once it complies with all outstanding technical requirements.

With kind regards,

Adrian J.F. Luty, PhD

Academic Editor

PLOS ONE

Additional Editor Comments (optional):

Following review both by myself and an expert reviewer, the revised manuscript is now deemed acceptable for publication.

Reviewers' comments:

Reviewer's Responses to Questions

**Comments to the Author**

1. If the authors have adequately addressed your comments raised in a previous round of review and you feel that this manuscript is now acceptable for publication, you may indicate that here to bypass the “Comments to the Author” section, enter your conflict of interest statement in the “Confidential to Editor” section, and submit your "Accept" recommendation.

Reviewer #1: (No Response)

2. Is the manuscript technically sound, and do the data support the conclusions?

Reviewer #1: (No Response)

3. Has the statistical analysis been performed appropriately and rigorously? 

Reviewer #1: (No Response)

4. Have the authors made all data underlying the findings in their manuscript fully available?

Reviewer #1: (No Response)

5. Is the manuscript presented in an intelligible fashion and written in standard English?

Reviewer #1: (No Response)

6. Review Comments to the Author

Reviewer #1: Many thanks to the authors for addressing my comments in their response and modifying the Results and Discussion section of the manuscript accordingly. This is now acceptable for publication as-is.

I have one final comment/question regarding the authors' definition of protection in their future studies. How do you prove evidence of 'exposure to recent infectious bites' unless you have conducted a controlled human malaria infection on these subjects and then dissect the mosquitoes?? Otherwise you can only assume that the subject was actually bitten. If you are basing this evidence on the presence of circulating blood-stage parasites (which does not necessarily imply 'recent' exposure, but that aside), then in practice your definition of protection equates to having asymptomatic parasitaemia. Although this is a potential indicator of clinical immunity/protection, it is a curious definition to use if you are investigating immune responses against a pre-erythrocytic antigen such as CSP. There you would presumably expect 'protection' to prevent the liver stage cycle from being completed, thus actually avoiding the appearance of blood-stage parasites - the precise opposite of your definition.

7. PLOS authors have the option to publish the peer review history of their article (what does this mean?). If published, this will include your full peer review and any attached files.

Reviewer #1: Yes: Matthew B.B. McCall

---

## [Editor Report · Acceptance letter]

15 Jan 2020

PONE-D-19-25608R1 

Identification of *Plasmodium falciparum* circumsporozoite protein-specific CD8+ T cell epitopes in a malaria exposed population 

Dear Dr. Kusi:

I am pleased to inform you that your manuscript has been deemed suitable for publication in PLOS ONE. Congratulations! Your manuscript is now with our production department. 

With kind regards,

on behalf of

Dr. Adrian J.F. Luty 

Academic Editor

PLOS ONE